# Surface Hardening of Massive Steel Products in the Low-pressure Glow Discharge Plasma

**Sergey Grigoriev, Alexander Metel \*, Marina Volosova, Yury Melnik, Htet A. Ney and Enver Mustafaev**

Department of High-efficiency Machining Technologies, Moscow State University of Technology "STANKIN", Vadkovsky per. 3A, 127055 Moscow, Russia

\* Correspondence: a.metel@stankin.ru; Tel.: +7-903-246-43-22

**Abstract:** A process vacuum chamber is filled with a homogeneous plasma of glow discharge with electrostatic electron confinement, which is used for surface hardening of massive products. At the current of 2–20 A and the gas pressure ranging from 0.1 to 1 Pa the discharge voltage amounts to 350–500 V. When a bias voltage of 2 kV is applied to an immersed in the plasma hollow cylinder with a mass of 15 kg, electrical power spent on heating it by accelerated ions exceeds by an order of magnitude the power spent on the discharge maintenance. The massive cylinder is heated up to 700 °C for 15 min. When argon mixture with nitrogen (30%) is used, the nitriding for 3h results in an increase in the surface hardness from 400 up to 1000 HV50 and the nitrided layer thickness grows to ~100 µm. The nitriding rate is enhanced by a high degree of nitrogen dissociation due to decomposition by fast electrons and surface structural defects due to bombardment by high-energy ions.

**Keywords:** glow discharge; electron confinement; plasma homogeneity; nitrogen dissociation; structural defects; nitriding rate

## 1. Introduction

Ion nitriding is widely used for hardening the surface of different machine parts [1,2]. It improves their corrosion [3–7], wear [8,9] and fatigue [10,11] resistance. The process includes introducing active nitrogen to the product surface and its subsequent diffusion into the bulk [12]. The nitrogen introduction and the product heating up to a temperature of effective nitrogen diffusion are carried out by means of the product bombardment by ions accelerated from the gas discharge plasma [13]. Increase in the energy of these ions results in growing the number of atoms knocked out of lattice sites. The surface structural defects facilitate the nitrogen introduction into the product surface. It is easier for smaller nitrogen atoms to penetrate through those defects into the surface layer compared to bigger nitrogen molecules. Hence, an increase in the degree of nitrogen dissociation in the plasma [14] also facilitates the nitrogen introduction into the surface. Furthermore, an increase in the product temperature accelerates the diffusion of introduced nitrogen atoms into the bulk.

The surface modification process traditionally makes use of abnormal glow discharge [15,16]. In contrast to the normal glow discharge, whose cathode is only partially covered with the so-called negative glow, in the abnormal discharge the whole surface of its cathode contacts with the discharge plasma and is separated from the plasma with a positive space charge sheath. The sheath width $d$ and the ion current density $j$ on the cathode depend on the discharge voltage $U$ and the gas pressure $p$. Distinctive features of this discharge are the values of $j/p^2 = F_1(U)$ and $pd = F_2(U)$ depending only on the discharge voltage $U$. For the discharge in nitrogen, these values have been first measured by Alexander Güntherschulze and published in his paper [17]. For more convenient use, we presented his data graphically in Figure 1.

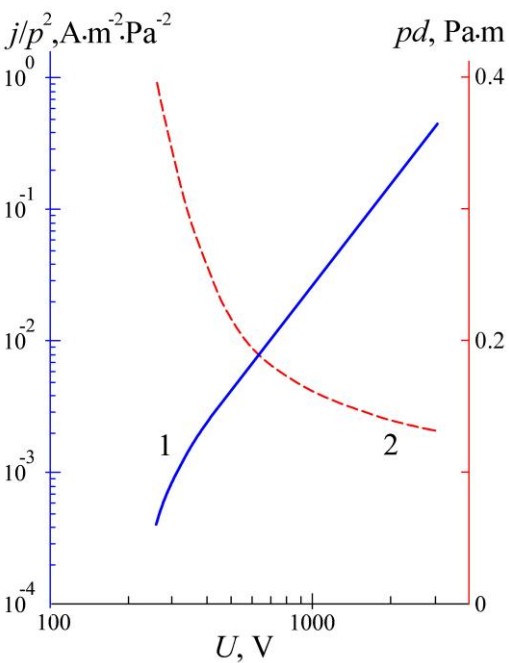

**Figure 1.** The ratio of the current density $j$ on the cathode of abnormal glow discharge in nitrogen to the square of the gas pressure $p$ ($j/p^2$, solid curve 1) and product of $p$ and the cathode sheath width $d$ ($pd$, dashed curve 2) versus discharge voltage $U$.

When a steel product to be nitrided in a vacuum chamber is fastened to feedthrough on the chamber top and connected to the negative pole of a power supply (Figure 2) the supply to the product of a negative voltage initiates an abnormal glow discharge between the chamber (anode) and the product (cathode).

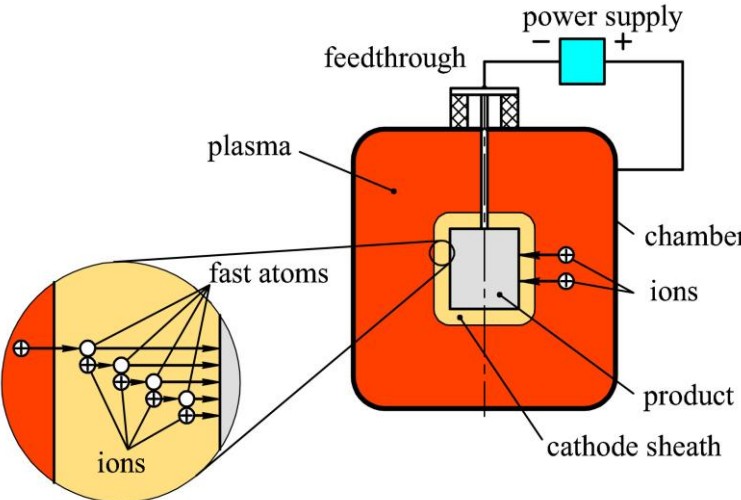

**Figure 2.** Schematic of plasma nitriding in an abnormal glow discharge.

The entire surface of the product is covered with a brightly glowing layer of the discharge negative glow, which is the nearest to the surface part of the discharge plasma. The plasma emits onto the product surface ions accelerated in the cathode sheath by the applied voltage. The ions bombard and heat the product. When the gas pressure is equal to 10 Pa the ion current density amounts to $j = 1$ A/m$^2$

and the heating power density on the product surface is equal to $w = 0.7$ kW/m$^2$. At a stationary temperature $T$ of a heated product, $w$ is equal to luminosity $R^*$ of its surface:

$$R^* = \omega \sigma_{\text{S-B}} T^4 \tag{1}$$

where $\omega$ is emissivity of the product surface and the Stefan–Boltzmann constant $\sigma_{\text{S-B}}$ is equal to $5.7 \times 10^{-8}$ W/(m$^2$·K$^4$). For a steel product $\omega \approx 0.25$ and at $R^* = w = 0.7$ kW/m$^2$ its temperature is equal to 198 °C, which is not enough for nitriding.

When the pressure rises to 50 Pa, the current density grows to $j = 25$ A/m$^2$, the heating power density on the product surface $w$ rises to 17.5 kW/m$^2$ and the temperature grows to 820 °C, which is quite enough for an effective nitriding. After the necessary temperature of the product is reached, it can be maintained by adjusting the heating power by changing the gas pressure or voltage.

At a pressure ~50 Pa and discharge voltage $U = 700$ V, an ion accelerated in the sheath collides with a gas atom at a distance of about 0.2 mm from the plasma border, takes an electron from the atom and turns it into a fast, neutral atom. The slow ion produced as a result of the electron transfer is also accelerated in the electric field of the sheath. After passing about the same distance of 0.2 mm it turns into a second fast neutral atom and a second slow ion appears. Through a 4-mm-wide sheath, the charge of one ion is transported by about 20 particles, which transfer the charge to each other. Their average energy amounts to 700/20 = 35 eV. At this energy, the influence of the surface structural defects is negligible, and the nitriding rate is determined only by the product temperature. For this reason, it takes sometimes 5–10 h to obtain nitrided layers of 100 μm in the abnormal glow discharge.

The nitriding time decreases to 1–2 h with the gas pressure reduction below 1 Pa. At such low pressure, the plasma can be produced by a high-frequency discharge [18], a vacuum arc [19], or a discharge with a thermionic cathode [20,21]. A product is immersed in the plasma and voltage of hundreds of volts is applied to it. At low pressures, ions from the plasma fly to the product without collisions in the space charge sheath between the plasma and the product and bombard its surface with the energy of hundreds of eV. The structural defects they create in the surface layer promote the introduction of nitrogen into the product and increase the nitriding rate.

It was shown in [18] that the nitriding rate in RF discharge at a pressure of ~0.5 Pa is appreciably higher than in the abnormal glow discharge at pressures of ~100 Pa. By means of the DC glow discharge assisted by thermionic emission (triode configuration), nitrided layers of 100–150 μm were obtained on high-speed steel substrates at the gas pressure of 0.5 Pa and the substrate temperature of 480–500 °C for 240 min, which indicates a substantial decrease in the nitriding time [20]. Under these conditions, each accelerated part from the plasma ion passes through the sheath to the product surface without collisions. It produces in the surface layer about ten structural defects, which enhance the nitriding process.

Unfortunately, it is impossible to use the triode glow discharge for production of homogeneous plasma in the chamber with a massive product loaded inside (Figure 2). It was attempted to nitride products in the vacuum-arc plasma [19]. An arc cathode of a coating deposition system was separated from the products loaded in the chamber center by a grounded screen shaped as jalousie. A voltage of ~20 V was applied to an additional electrode introduced into the chamber. Metal droplets emitted by the cathode spots, metal ions and atoms were deposited on the screen. Despite the plasma purified in this way from metal particles is highly non-homogeneous, it was successfully used for cutting tools pre-nitriding on planetary rotation system before deposition of wear-resistant coatings. For plasma generation at low gas pressure and plasma nitriding of the substrates can be also used electron beams [22–24].

Surface modification processes influence microstructure and properties of steels [25–29] even at a low-temperature nitriding process [30]. Hydrogen-free ion nitriding improves the mechanical characteristics and performance of hard alloys T5K10 and T15K6 [31]. Combined processing composed of plasma nitriding and wear-resistant coating deposition ensures a perfect tribological behavior

of steel [32]. Plasma nitriding can also improve the adhesion of DLC films deposited on steel by magnetron sputtering [33].

A steel surface can be also hardened using plasma immersion ion implantation [34]. In this case, high-voltage pulses (10–100 kV) are applied to a sample immersed in a nitrogen plasma. Accelerated ions bombard the sample, and nitrogen atoms penetrate to a depth of 50–500 nm. Width of a sheath between the plasma and the sample can exceed 10–20 cm [35]. Therefore, this method cannot be used for nitriding the inner surface of hollow products, for instance, a cylinder with inner diameter not exceeding 10 cm, and the chamber sizes should amount to ~0.5 m for small samples and to ~1 m for bigger products. To prevent accelerated ions from energy losses due to collisions in the sheath with molecules, the gas pressure should be less than 0.1 Pa.

In [36] high voltage pulses between 5 and 30 kV were applied to small samples made of stainless steel, and it was found that heating the samples allows an increase in nitrided layer thickness from 0.3 to 2 μm. A rf plasma source at the gas pressure of 0.3 Pa generated a plasma composed of approximately 95% $N_2^+$ and 5% $N^+$. Its power (150 W) and power of the high-voltage pulse generator were not enough, and an additional heating system consisting of IR lamps had to be used to achieve sample temperatures of 600 °C and beyond. Comparatively low degree of nitrogen dissociation ~5% cannot enhance the nitriding process. It is obvious that the above ion implantation system is not suitable for the production of 100-μm-thick nitrided layers on the surface of massive steel products.

The goal of the present research is the development of surface hardening technology, which could increase the nitriding rate of the massive products due to bombardment by high-energy ions and a high degree of nitrogen dissociation.

## 2. Materials and Methods

To achieve the goal, it was necessary to fill the process chamber presented in Figure 2 with a homogeneous plasma produced by a gas discharge at a pressure of $p = 0.1$–1 Pa. It was shown in [37], that due to hollow cathode effect based on the multiplication of fast electrons in the cathode sheath of glow discharge, quite homogeneous plasma can be produced at p = 0.01–1 Pa. It has been already used as a plasma emitter of ions [38,39].

Figure 3 shows an experimental setup based on plasma generation using the hollow cathode glow discharge. Here the process chamber plays the role of the hollow cathode. The 85-cm-high chamber with an inner surface area $S_{ch} = 2.5$ m$^2$ and volume of $V = 0.26$ m$^3$ is shaped like a hexagonal prism with the incircle diameter of 60 cm. To high-voltage feedthrough mounted on the top of the chamber can be fastened a 20-cm-long hollow cylinder made of AISI 5135 steel with an external diameter of 30 cm and the inner diameter of 28 cm. A flat anode with a variable surface area is fastened to another feedthrough.

The setup is equipped with two DC power supplies mounted in a rack. One of them with stabilized current up to 20 A at voltages of $U_d = 200$–500 V is connected between the anode (positive pole) and the chamber (negative pole). It is used as a discharge power supply. The second power supply with stabilized voltage up to 2 kV at current up to 10 A is connected between the high-voltage feedthrough (negative pole) and the chamber (positive pole). It is used as a bias voltage power supply.

The chamber is evacuated by a turbo-molecular pump, which ensures the residual gas pressure of 0.001 Pa. The operating gas pressure is regulated from 0.01 to 5 Pa by a two-channel gas supply system. For monitoring, control and characterization of ions, neutrals, and radicals in the discharge plasma the chamber is equipped with a high performance combined mass and energy quadrupole mass spectrometer for the characterization of plasmas Hiden EQP (300 AMU mass range and 1000 eV energy range) produced by Hiden Analytical ltd (England).

A movable disc probe (not shown in Figure 3), which is negatively biased to 40 V relative to the anode is used for measuring the ion density distribution inside the chamber. The temperature of the cylinder and other substrates immersed in the plasma is measured through a quartz window on the

chamber wall by pyrometer IMPAC IP 140 produced by LumaSense Technologies GmbH (Germany). All systems of the setup are monitored using a control panel.

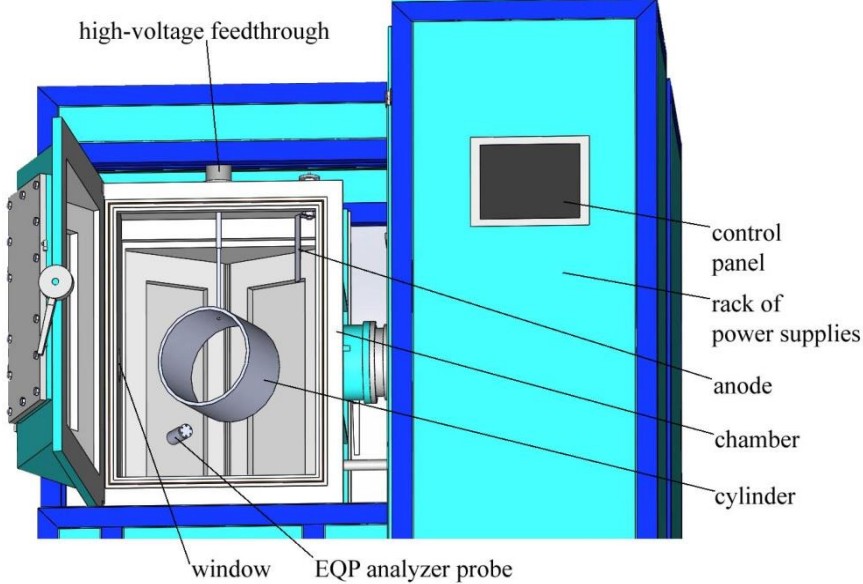

**Figure 3.** Experimental setup with an open door of the vacuum chamber.

## 3. Results

### 3.1. Generation of Homogeneous Plasma in a Big Chamber Volume

At the gas pressure of 0.5 Pa, activation of the discharge power supply results in the glow discharge ignition. The chamber is filled with a homogeneous glow of the discharge plasma (Figure 4). At the gas pressure ranging from 0.1 to 1 Pa the discharge voltage $U_d$ grows with current $I_d$ and amounts to 350–500 V at $I_d$ of 2–20 A.

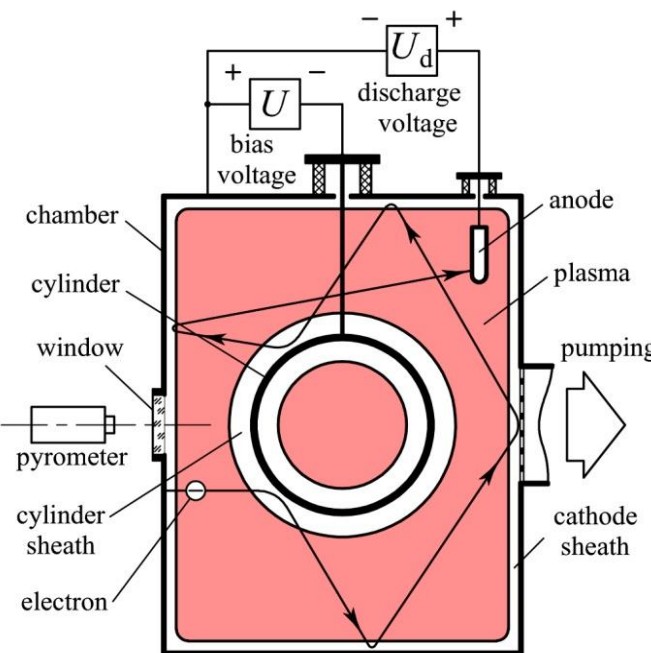

**Figure 4.** Schematic of the homogeneous plasma production by a glow discharge with electrostatic confinement of electrons.

Emitted by the chamber electrons are accelerated in the cathode sheath of positive space charge between the chamber wall and the plasma up to the corresponding energy of 350–500 eV. They pass through the plasma to the opposite chamber wall or the cylinder immersed in the plasma and are reflected back by the electric field in the opposite sheath. Due to multiple reflections, each fast electron can visit all parts of the chamber volume and ionize the gas everywhere. This ensures the plasma homogeneity.

The broken trajectory length of the electron way from the chamber wall to the anode can exceed the chamber sizes by two orders of magnitude. For this reason, electrons can spend all their energy on the gas ionization at record low pressures. This ensures the maintenance of the discharge. In this case, emitted by the chamber electron produces inside the chamber $N = eU_d/W$ new free electrons, where $W$ is the gas ionization cost [40]. For instance, the ionization cost of argon is equal to 26 eV.

In order to spend all their initial energy on the gas excitation and ionization those electrons have to pass in the gas a way $\Lambda = N\lambda_N = (eU_d/W)\lambda_N$. Here the ionization length $\lambda_N$ is their mean free path between ionizing collisions $\lambda = 1/n_o\sigma$ averaged over the whole range of their energy decreasing from $eU_c$ down to $W$, where $n_o$ and $\sigma$ are the gas atoms density and the ionization cross-section. At argon pressure $p = 0.1$ Pa and discharge voltage $U_d = 416$ V the number of ionizations amounts to $416/26 = 16$, $\lambda_N = 1.6$ m and $\Lambda = 25.6$ m.

The chamber walls with the surface area of $S_{ch} = 2.5$ m$^2$ and the cylinder walls with the surface area of $S_c = 0.38$ m$^2$ form an electrostatic trap for the electrons oscillating inside. The trap volume is equal to $V = 0.26$ m$^3$ and its surface amounts to $S = S_{ch} + S_c = 2.88$ m$^2$. Due to multiple reflections from the walls, electrons always change the direction of their movement and may be assumed to be isotropic. The only way out of the trap is the absorption of the electrons by the anode with the surface area $S_a$. The length $L$ of the way from the cathode to the anode of emitted by the cathode electrons with isotropic velocity distribution amounts [38] to

$$L = 4V/S_a \qquad (2)$$

At $p = 0.1$ Pa this length $L$ is equal to $\Lambda = 25.6$ m when the anode surface area $S_a = 4V/L = 4 \times 0.26/25.6 \approx 0.04$ m$^2$. At this anode surface area $S_a = 0.04$ m$^2$, a constant discharge current $I_d = 10$ A and argon pressure $p > 0.1$ Pa the discharge voltage $U_d$ is independent of $p$ and amounts to $U_d = 390$ V. When $p$ becomes lower than 0.1 Pa the discharge voltage $U_d$ starts rising with a further decrease in the pressure and grows to $U_d = 490$ V at $p = 0.05$ Pa (Figure 5).

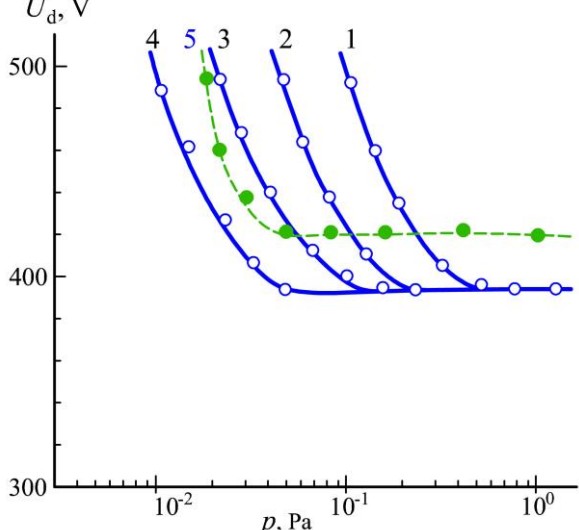

**Figure 5.** Dependence of the discharge voltage $U_d$ on argon pressure $p$ at discharge current $I_d = 10$ A and the anode surface area $S_a = 0.16$ (1), 0.08 (2), 0.04 (3), 0.02 (4) and 0.01 m$^2$ (5).

An increase in the anode surface area up to $S_a = 0.08$ m$^2$ results in growth by two times from 0.1 to 0.2 Pa of the pressure $p_o$, at which the discharge voltage $U_d$ starts rising. Further increases in $S_a$ up to 0.16 m$^2$ led to growth in the pressure $p_o$ up to 0.4 Pa. The decrease in the anode surface area $S_a$ from 0.04 to 0.02 m$^2$ resulted in $p_o$ reduction by two times from 0.1 to 0.05 Pa. It can be concluded that the pressure $p_o$ is directly proportional to the anode surface area $S_a$. Hence, it could be predicted that using an anode with surface area $S_a = 0.01$ m$^2$ the pressure $p_o$ will be reduced to 0.025 Pa.

However, at the anode surface $S_a = 0.01$ m$^2$ the discharge voltage starts rising at a pressure of ~0.06 Pa and at higher pressures it exceeds by 10–20 V the discharge voltage at $S_a = 0.02$–0.16 m$^2$. In this case, brightly glowing plasma was observed on the anode surface. Presence of the anode plasma can be explained with positive anode fall of potential, which appears when the anode surface area $S_a$ is less than a critical value:

$$S^* = (\pi/e)^{1/2}(2m/M)^{1/2}S \approx (2m/M)^{1/2}S \tag{3}$$

where e is the Naperian base, $m$ and $M$ are the electron mass and the ion mass, and $S$ is the cathode surface area [41]. In our case $S = S_{ch} = 2.5$ m$^2$, electron mass $m = 9.1 \times 10^{-31}$ kg, argon ion mass M = 40 $\times 1.66 \times 10^{-27}$ kg $= 6.64 \times 10^{-26}$ kg and $S^* = 0.013$ m$^2$. The anode surface area $S_a = 0.01$ m$^2$ (dashed curve 5 in Figure 5) is really less than the critical value $S^*$, which results in the positive anode fall of potential.

With decreasing gas pressure, the anode fall of potential grows from tens to hundreds of volts. At a limited value of the discharge voltage, it results in a reduction of the cathode fall of potential and extinguishing of the discharge. Therefore, special attention should be paid to the anode used for plasma production in a process chamber so that its surface area could slightly exceed the critical value $S^*$ dependent on the chamber sizes and the sort of the gas.

Figure 6 presents spatial distributions of the ion current from the discharge plasma to the movable probe. The ion current density is proportional to the plasma density, and the distributions characterize the plasma as quite homogeneous. For instance, nonuniformity of the plasma in the central part of the chamber does not exceed 10% at the discharge current $I_d = 20$ A (curve 2).

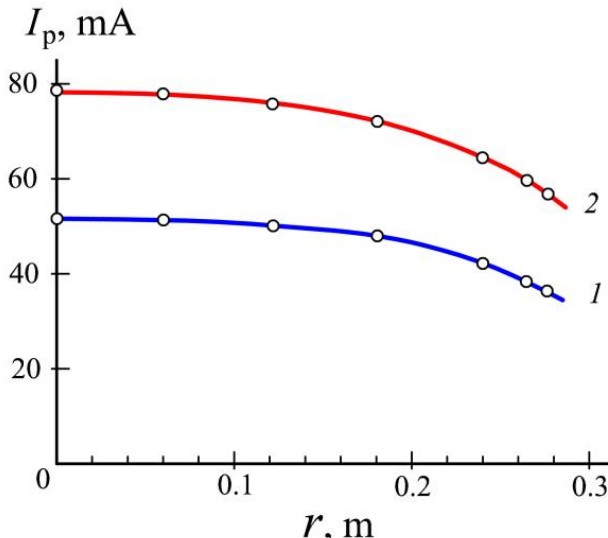

**Figure 6.** The probe current $I_p$ versus distance $r$ from the chamber axis at $I_d = 12$ (1) and 20 A (2).

When a cylinder is immersed in the plasma (Figure 4) and at the same current $I_d = 20$ A in the anode circuit a negative bias voltage of $U = 2$ kV is applied to the cylinder the current in its circuit $I$ grows from 3 to 9 A and the discharge voltage $U_d$ diminishes from 440 to 150 V. Potential difference between the plasma and the cylinder is equal to $U+U_d$ and the cylinder is bombarded by ions with energy of 2.15 keV. At this energy coefficient of secondary ion-electron emission amounts to $\gamma_i \sim 0.5$. It means that one-third of the current $I = 9$ A in the cylinder circuit is the current of electrons emitted by

the cylinder surface of $S_c$ = 0.4 m$^2$, which bombard the chamber walls with the energy of 2 keV. At this energy coefficient of secondary electron-electron emission amounts to $\gamma_e$ ~ 1. Hence, the electron bombardment gives rise to electron emission current of 3 A from the chamber walls.

At the ion energy 150 eV $\gamma_i$ ~ 0.1 and at the current in the chamber circuit of $I_d - I = 20 - 9 = 11$ A the electron emission current of 3 A due to electron bombardment sufficiently exceeds the emission current of 1.1 A due to ion bombardment. Increase in the emission current of the chamber is the reason for a decrease in discharge voltage after application of high voltage to the cylinder.

It should be mentioned that in spite of the appreciable decrease in the discharge voltage $U_d$, the immersion of the cylinder in the plasma and application to the cylinder of high voltage only slightly influences the plasma density distribution. For instance, the distribution along the cylinder axis at the current in the anode circuit $I_d$ = 20 A practically does not differ from the distribution obtained without cylinder (curve 2 in Figure 6).

Figure 7 presents energy distributions of nitrogen ions, which bombard the chamber walls and the grounded end cap of the Hiden EQP analyzer probe (see Figure 3). Ions are extracted from the plasma produced in the mixture of argon with nitrogen (30%) at the gas pressure of 0.9 Pa. They are accelerated by the discharge voltage $U_d$ and enter the probe through a 50-µm-diameter orifice in the center of the end cap. Their maximal energy exactly corresponds to the discharge voltage. The distributions show that the current densities of atomic and molecular nitrogen ions both grow with an increase in the discharge current $I_d$. However, the ratio of the atomic ion current density to the molecular ion current density rises from 0.8 at $I_d$ = 5 A to 0.95 at $I_d$ = 10 A and to 1.4 at $I_d$ = 20 A. It means that at $I_d$ = 20 A the degree of nitrogen dissociation exceeds 50%. The reason is a great number of fast electrons oscillating inside the chamber, which collide with the nitrogen molecules and decompose them. We may suppose that the ratio of the atomic to molecular nitrogen ion current densities on the surface of the negatively biased cylinder is about the same as on the chamber walls.

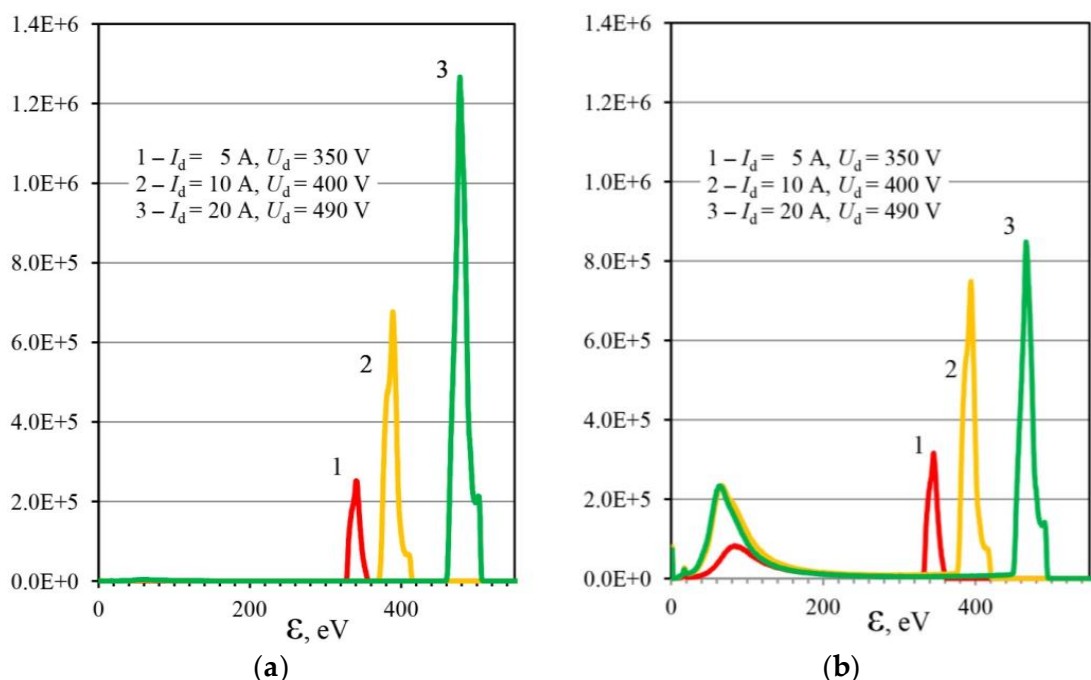

**Figure 7.** Energy spectra of atomic (**a**) and molecular (**b**) nitrogen ions arriving at the chamber walls at the gas pressure of 0.9 Pa and the discharge current $I_d$ = 5 (1), 10 (2) and 20 A (3).

Figure 7b reveals a group of molecular ions with a mean energy of 60–80 eV. They are due to the charge exchange collisions in the sheath between the plasma and the end cap of the Hiden EQP analyzer probe. The number of those ions is proportional to the gas pressure. When the pressure is

reduced from 0.9 to 0.1 Pa molecular nitrogen ions with energy not exceeding 200 eV disappear from the energy spectrum.

In the range from 0.1 to 1 Pa no low energy atomic ions were observed because the charge exchange cross-section for atomic nitrogen ions is much less than the cross-section of resonance charge exchange collisions for molecular nitrogen ions [42,43]. Bombardment of a product surface by atomic ions instead of molecular ions facilitates the introduction of nitrogen into the product surface and greatly contributes to an increase in the nitriding rate. Effective decomposition of gas molecules by fast electrons in the hollow-cathode glow discharge is its well-known distinctive feature [44].

### 3.2. Plasma Nitriding at a Low Gas Pressure

Maximal electrical power spent on the cylinder heating 2.15kV × 6A = 13 kW appreciably exceeds the power spent on the maintenance of the discharge 0.15kV × 11A = 1.65 kW. Temperature measurements using pyrometer IMPAC IP 140 (LumaSense Technologies GmbH, Germany) showed that in argon plasma it takes only 15 min to heat to 700°C a steel cylinder with a mass of 15 kg. After the given temperature is reached the discharge current and/or the bias voltage on the cylinder are diminished and the temperature is stabilized by regulation of the discharge current.

In order to assess the heating uniformity, the temperature was measured in marked points on the outer surface of the cylinder distanced from one of its ends at 1, 3, 5, 7 and 9 cm. Presented in Figure 8 results show that the temperature is distributed quite homogeneously. A slight increase in the temperature near the cylinder ends is due to ion current density on the ends being higher than in the middle of the cylinder.

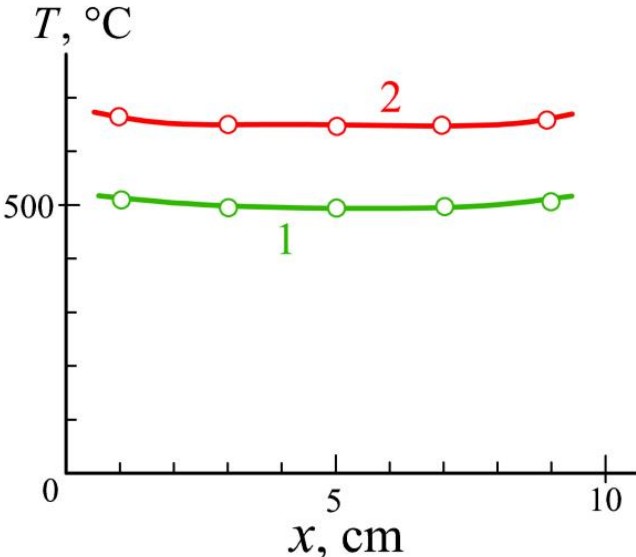

**Figure 8.** Dependence of the cylinder surface temperature $T$ on the distance $x$ from one of its ends at the bias voltage $U = 2$ kV and currents in the cylinder circuit $I = 1.5$ (1) and 3A (2).

Nitriding of the cylinder was carried out for 180 min in argon mixed with nitrogen (30%) at the temperature ~560°C and the bias voltage of −2000 V. To assess the processing results, five 15-mm-long 10-mm-wide and 2-mm-thick substrates were placed inside and outside the cylinder. The substrates were made of the same material (AISI 5135). Three of them were placed inside the cylinder and distanced from its end at 1, 3 and 5 cm. Two others were placed outside on the cylinder top and distanced from the cylinder end at 1 and 3 cm.

After nitriding, the hardness and depth of the nitrided layers of the substrates were evaluated using Vickers microhardness measurements on polished cross-sections of the substrates under a load

of 50 g. Presented in Figure 9, distributions show that the nitrided layers depth amounts to about 100 µm. The hardness of the surface is about 2.5 times higher than the hardness of the bulk.

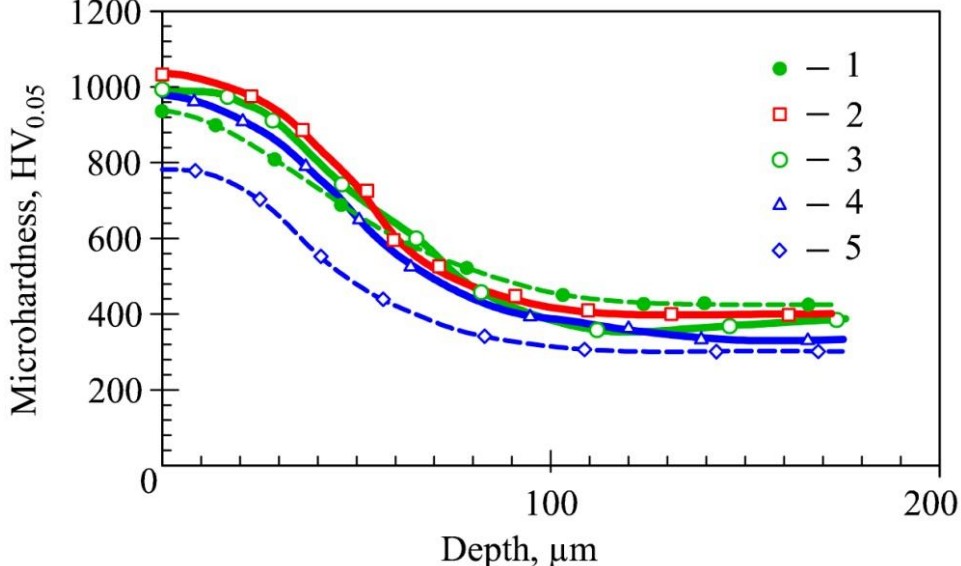

**Figure 9.** Microhardness versus distance from the surface of substrates placed outside the cylinder at 3 cm from its end (1) and inside the cylinder at 1 cm (2), 3 cm (3) and 5 cm (4) from its end and nitrided at the bias voltage $U = -2000$ V. Curve 5 presents distribution for the substrate nitrided at $U = -500$ V.

As the substrates placed on the inner surface of the hollow cylinder were bombarded by ions with the same energy and current density at the same temperature of 560 °C, we may suppose that the nitrided layer on the whole surface of the cylinder has about the same characteristics.

After those experiments, another substrate was placed inside the cylinder at a distance of 5 cm from its end. It was nitrided for 180 min in argon mixed with nitrogen (30%) at the same temperature ~560 °C but at a lower bias voltage of −500 V. In order to keep the processing temperature, the discharge current in the anode circuit was increased. The microhardness distribution presented by curve 5 in Figure 9 demonstrates, in this case, an appreciable decrease in the nitrided layer thickness and microhardness values. As the temperature was kept the same, those changes can be related only to a decrease in the density of surface structural defects, which promote introducing of the active nitrogen to the product and contribute to increase in the nitriding rate.

For evaluation of the cylinder wear-resistance, its surface before the nitriding was treated using a dry injector blasting installation 75 S produced by Sandmaster AG (Switzerland). Five cavities were produced on the surface by a blasting pistol with a 4-mm-diameter hard metal nozzle for $t = 1, 2, 3, 4$ and 5 min. Depth of the cavity $h$ was measured using the optical measuring system MicroCAD premium+ by GFMesstechnik GmbH (Germany). Figure 10 shows that the wear of the cylinder surface monotonically grows with the blasting time. The depth of the cavity rises from $h = 105$ µm at $t = 1$ min to $h = 210$ µm at $t = 5$ min. When the nitrided cylinder was treated the surface wear after 1-min-long blasting was about two times lower than for the untreated surface and depth of the cavity amounted to $h = 60$ µm. The depth slowly increased to $h = 90$ µm at $t = 4$ min. However, after 5-min-long blasting, the cavity depth was two times higher. This indicates that at $t > 4$ min, the wear-resistant nitrided layer was already destroyed.

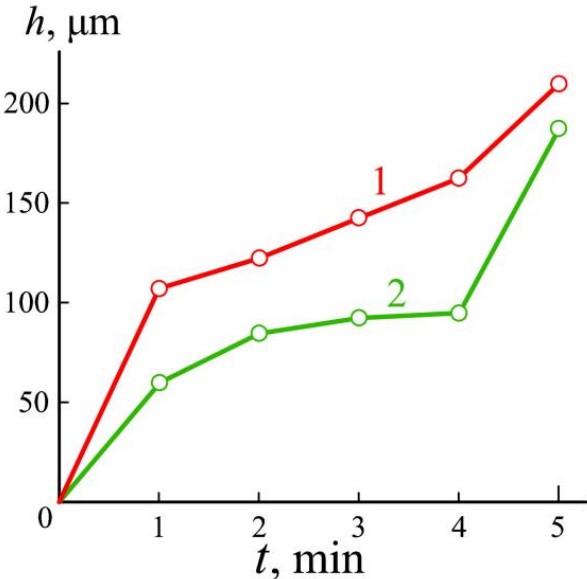

**Figure 10.** Depth of the cavity h versus cylinder blasting time t before (1) and after (2) nitriding.

## 4. Discussion

The results obtained reveal new technological capabilities of the glow discharge with electrostatic electron confinement. The discharge can fill any process vacuum chamber with quite homogeneous plasma at the gas pressure of 0.1–1 Pa and ensures uniform heating of massive products. Therefore, the discharge plasma can be used for the heat treatment of the products and their surface hardening.

The discharge is remarkable for a great number of fast electrons oscillating inside the process vacuum chamber. They not only allow the maintenance of the glow discharge at record low gas pressures but also promote decomposition of nitrogen molecules in the discharge plasma. The investigation results showed that the dissociation degree of nitrogen can exceed 50%. As it is easier for smaller nitrogen atoms to penetrate into the product surface, the production of active nitrogen facilitates introducing it to the product and contributes to an increase in the nitriding rate.

Low gas pressure allows an increase in the energy of ions accelerated from the gas discharge plasma, which produce structural defects on the product surface. Those defects also promote the introduction of active nitrogen to the product and contribute to an increase in the nitriding rate.

Independent regulation of the bias voltage allows the product bombardment by ions with high energy exceeding 1 keV. Structural defects induced by those ions and a high degree of nitrogen dissociation by fast electrons appreciably increase the nitriding rate compared to traditional ion nitriding in the abnormal glow discharge. The product heating power can, by an order of magnitude, exceed the power spent on discharge maintenance.

Comparison of the new ion nitriding system with the traditional one allows a future research direction—modification of available ion nitriding systems based on abnormal glow discharge. It is only necessary to introduce an anode into the vacuum chamber of the system and to connect between them a DC power supply ensuring a stabilized discharge current of 10–20 A.

**Author Contributions:** Conceptualization, A.S.M., S.N.G.; methodology, M.A.V., Y.A.M.; software, E.S.M.; validation, H.A.N.; formal analysis, S.N.G.; investigation, A.S.M., Y.A.M, H.A.N., E.S.M.; resources, S.N.G., M.A.V.; data curation, H.A.N.; writing—original draft preparation, M.A.V.; writing—review and editing, E.S.M.; visualization, Y.A.M. and A.S.M.; supervision, S.N.G.; project administration, M.A.V.; funding acquisition, S.N.G.

**Funding:** This research was funded by the Ministry of Education and Science of the Russian Federation in the framework of the state task in the field of scientific activity of Moscow State University of Technology "STANKIN", grant number 9.7886.2017/6.7.

**Acknowledgments:** The work was carried out using the equipment of the Center of collective use of MSUT "STANKIN".

**Conflicts of Interest:** The authors declare no conflict of interest.

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
