# Peer review of "Surface Hardening of Massive Steel Products in the Low-pressure Glow Discharge Plasma"

_technologies, doi:10.3390/technologies7030062_

Round 1

Reviewer 1 Report

The manuscript reports interesting results about a process which allows efficient massive steel nitriding owing to high energy ions and high degree of dissociation of nitrogen due to electrostatic electron confinement in a glow discharge at low pressure. The topic is quite relevant of many journal's readers.

However, the text needs great improvement, especially the english. The abstract must be thoroughly revised. Some suggestions are indicated below.

Moreover, one major revision must be considered before the paper be published in the journal.

The authors should compare their results with those obtained under conditions of unpolarized cylinder to check the efficiency of their process.

They also should compare their process and results with those previously obtained by ion implantation process, plasma immersion ion implantation. See for example: G.A. Emmert et al J.Appl.Phys. 71 (1) 113-117 (1992), J.R. Conrad et al Surf. Coat. Technol. 36, 927 (1988), S. Mändl at al. 200, 584-588 (2005).

As a general rule, it would be fine, an english native or people speaking english well could revise the manuscript.

Some suggestions: line 91," Unfortunatly" instead of "regrettably", "under condition"s instead of "in conditions", "at the gas pressure" replaced by "at a gas pressure", "with decreasing gas pressure" instead of 'with decreasing of the..." (line 191 and throughout the text), line 131 "They are produced by" or "They are due to", Fig 9 and everywhere "versus distance" instead of "versus the distance". Put comma to separate various groups of words in sentences.

Product ? rather sample or steel sample or specimen or steel specimen.

Author Response

Response to Reviewer 1 Comments

Point 1: The manuscript reports interesting results about a process which allows efficient massive steel nitriding owing to high energy ions and high degree of dissociation of nitrogen due to electrostatic electron confinement in a glow discharge at low pressure. The topic is quite relevant of many journal's readers. However, the text needs great improvement, especially the english. The abstract must be thoroughly revised. Some suggestions are indicated below.

Response 1: Thank you for your positive appreciation of our work and justified suggestions to improve English. We used for this purpose a special program “Grammarly”. And of course, we agreed with you and replaced "regrettably" by "unfortunately", "in conditions" by "under conditions", "at the gas pressure" by "at a gas pressure", "with decreasing of the gas pressure" by "with decreasing gas pressure", "versus the distance" by "versus distance" and so on. At your request we have thoroughly revised the abstract.

Point 2: The authors should compare their results with those obtained under conditions of unpolarized cylinder to check the efficiency of their process.

Response 2: According to your advice, we compared the cylinder nitriding results at the bias voltage of 2000 V with nitriding at the same temperature but at a lower bias voltage of 500 V. Thereby we modified Figure 9. Additional curve 5 in Figure 9 shows that in the latter case an appreciable decrease in the nitrided layer thickness and microhardness values takes place. Those changes can be related only to a decrease in the density of induced by high-energy ions surface structural defects, which promote introducing of the active nitrogen to the product and hence contribute to an increase in the nitriding rate. This proves the efficiency of the nitriding process enhanced by high-energy ions.

Point 3: The authors also should compare their process and results with those previously obtained by ion implantation process, plasma immersion ion implantation. See for example: G.A. Emmert et al J.Appl.Phys. 71 (1) 113-117 (1992), J.R. Conrad et al Surf. Coat. Technol. 36, 927 (1988), S. Mändl at al. 200, 584-588 (2005)

Response 3: We also compared our results with those previously obtained by ion implantation process, plasma immersion ion implantation: G.A. Emmert et al J.Appl.Phys. 71 (1) 113-117 (1992); J.R. Conrad et al Surf. Coat. Technol. 36, 927 (1988); S. Mändl et al 200, 584-588 (2005).

Thereby we added those papers in the list of references [34-36].

In [34] high-voltage pulses (10–100 kV) are applied to a sample immersed in a nitrogen plasma. Accelerated ions bombard the sample, and nitrogen atoms penetrate to a depth of 50–500 nm. Width of a sheath between the plasma and the sample can exceed 10–20 cm [35]. Therefore this method cannot be used for nitriding the inner surface of hollow products, for instance, a cylinder with inner diameter not exceeding 10 cm.

Also the chamber sizes should amount to ~ 0.5 m for small samples and to ~ 1 m for bigger products. To prevent accelerated ions from energy losses due to collisions with molecules in the sheath the gas pressure should be less than 0.1 Pa.

In [36] high voltage pulses between 5 and 30 kV are applied to small samples made of stainless steel, and the samples heating allows an increase in nitrided layer thickness from 0.3 to 2 μm. An rf plasma source at the gas pressure of 0.3 Pa generates a plasma composed of approximately 95% N2+ and 5% N+. The source power (150 W) and power of the high-voltage pulse generator are quite low, and an additional heating system consisting of IR lamps is used to achieve sample temperatures of 600oC and beyond.

Comparatively low degree of nitrogen dissociation ~ 5% cannot enhance the nitriding process. It is obvious that such ion implantation systems are not suitable for the production of 100-μm-thick nitrided layers on the surface of massive steel products.

After scrutinizing the above papers we found no direct connection with the topic of our manuscript – surface hardening of massive steel products. In our case, the main hardening process is thermodiffusion of nitrogen enhanced by a high degree of nitrogen dissociation in plasma by fast electrons and surface structural defects induced by high-energy ions.

The surface layers hardened by nitriding are much thicker than those hardened by ion implantation. The latter should be better used for surface hardening of heat-sensitive materials where it has no rivals.

Reviewer 2 Report

This paper has many valuable results. However, the authors are requested to modify the style of presentation. For example, the Introduction section needs to provide a 'higher level' overview of plasma nitriding and not details about the process. The authors use specific numbers and parameters (e.g. see lines 62 - 90) whereas it would be better to talk in abstract concepts and save the numbers for the Results and Discussion section.

Author Response

Point 1: This paper has many valuable results. However, the authors are requested to modify the style of presentation. For example, the Introduction section needs to provide a 'higher level' overview of plasma nitriding and not details about the process.

Response 1: Thank you for your positive appreciation of our results and valuable advices to modify the style of presentation.

Point 2: The authors use specific numbers and parameters (e.g. see lines 62 - 90) whereas it would be better to talk in abstract concepts and save the numbers for the Results and Discussion section.

Response 2: According to your advice, we transferred most of specific numbers and parameters from the Introduction section to the Results and Discussion section.

Reviewer 3 Report

After analyzing the article, in my opinion, it is well thought out and ready to print.

However, one temperature unit should be used.

Author Response

Point 1: After analyzing the article, in my opinion, it is well thought out and ready to print.

Response 1: Thank you for your positive appreciation of our work.

Point 2: However, one temperature unit should be used.

Response 2: According to your advice, we left in the text only one temperature unit.

Round 2

Reviewer 1 Report

Your paper has been greatly improved. However, I still found english mistakes. I'm sure, you can revised your paper by reading it very carefully.

Good luck